# RELA∙8-Oxoguanine DNA Glycosylase1 Is an Epigenetic Regulatory Complex Coordinating the Hexosamine Biosynthetic Pathway in RSV Infection

**DOI:** 10.3390/cells11142210

**Published:** 2022-07-15

**Authors:** Xiaofang Xu, Dianhua Qiao, Lang Pan, Istvan Boldogh, Yingxin Zhao, Allan R. Brasier

**Affiliations:** 1Department of Medicine, School of Medicine and Public Health (SMPH), University of Wisconsin-Madison, Madison, WI 53705, USA; xxu@wisc.edu (X.X.); dqiao@wisc.edu (D.Q.); 2Department of Microbiology and Immunology, University of Texas Medical Branch, Galveston, TX 77555, USA; lapan@utmb.edu (L.P.); sboldogh@utmb.edu (I.B.); 3Department of Medicine, University of Texas Medical Branch, Galveston, TX 77555, USA; yizhao@utmb.edu; 4Institute for Clinical and Translational Research (ICTR), University of Wisconsin-Madison, Madison, WI 53705, USA

**Keywords:** RSV, hexosamine biosynthetic pathway, GFPT2, innate immunity, RELA, DNA-damage repair, 8-oxoguanine DNA glycosylase1

## Abstract

Respiratory syncytial virus (RSV), or human orthopneumovirus, is a negative-sense RNA virus that is the causative agent of severe lower respiratory tract infections in children and is associated with exacerbations of adult lung disease. The mechanisms how severe and/or repetitive virus infections cause declines in pulmonary capacity are not fully understood. We have recently discovered that viral replication triggers epithelial plasticity and metabolic reprogramming involving the hexosamine biosynthetic pathway (HBP). In this study, we examine the relationship between viral induced innate inflammation and the activation of hexosamine biosynthesis in small airway epithelial cells. We observe that RSV induces ~2-fold accumulation of intracellular UDP-GlcNAc, the end-product of the HBP and the obligate substrate of N glycosylation. Using two different silencing approaches, we observe that RSV replication activates the HBP pathway in a manner dependent on the RELA proto-oncogene (65 kDa subunit). To better understand the effect of RSV on the cellular N glycoproteome, and its RELA dependence, we conduct affinity enriched LC-MS profiling in wild-type and RELA-silenced cells. We find that RSV induces the accumulation of 171 N glycosylated peptides in a RELA-dependent manner; these proteins are functionally enriched in integrins and basal lamina formation. To elaborate this mechanism of HBP expression, we demonstrate that RSV infection coordinately induces the HBP pathway enzymes in a manner requiring RELA; these genes include Glutamine-Fructose-6-Phosphate Transaminase 1 (GFPT)-1/2, Glucosamine-Phosphate N-Acetyltransferase (GNPNAT)-1, phosphoglucomutase (PGM)-3 and UDP-N-Acetylglucosamine Pyrophosphorylase (UAP)-1. Using small-molecule inhibitor(s) of 8-oxoguanine DNA glycosylase1 (OGG1), we observe that OGG1 is also required for the expression of HBP pathway. In proximity ligation assays, RSV induces the formation of a nuclear and mitochondrial RELA∙OGG1 complex. In co-immunoprecipitaton (IP) experiments, we discover that RSV induces Ser 536-phosphorylated RELA to complex with OGG1. Chromatin IP experiments demonstrate a major role of OGG1 in supporting the recruitment of RELA and phosphorylated RNA Pol II to the HBP pathway genes. We conclude that the RELA∙OGG1 complex is an epigenetic regulator mediating metabolic reprogramming and N glycoprotein modifications of integrins in response to RSV. These findings have implications for viral-induced adaptive epithelial responses.

## 1. Introduction

Respiratory syncytial virus (RSV; human orthopneumovirus) is responsible for seasonal outbreaks of respiratory tract infections worldwide [1]. Consequently, RSV is the most-common cause of hospitalization in children 0–5 years of age [2,3]. Severe lower respiratory tract infections (LRTIs) with RSV results in giant cell formation, epithelial sloughing and necrosis, pathological events that produce mucous plugging, ventilation–perfusion mismatching, and hypoxic respiratory failure [4]. Prospective observational studies of children with severe RSV infections show that LRTIs are associated with long-term decreased pulmonary function [5,6].

Naturally acquired infections produce limited protective immunity. Consequently, RSV infections occur throughout life. In older adults with waning immune defenses, RSV can produce increasingly severe disease, such as pneumonia and wheezing exacerbations [7]. In patients with allergic asthma, viral infections, including RSV, are responsible for the majority of actute exacerbations [8]. Of concern, repetitive exacerbations have been linked to decrements in pulmonary function [9,10,11].

The mechanisms of how viral infection-induced airway inflammation are linked to structural remodeling are underexplored. By contrast, mechanisms of acute infections are better understood. RSV is transmitted by large droplet spread, fomites, or self-inoculation. Virus initially binds and replicates in ciliated airway epithelial cells in the upper nasopharynx, spreading into the lower small bronchiolar epithelium causing epithelial injury and persistent mucosal inflammation [4,12]. Preclinical studies using the tissue-selective genetic knockout of innate signaling in SAECs has implicated the lower airway epithelial cell in pathogenic inflammatory response and airway obstruction, the most prominent manifestations of human disease [13].

RSV replication activates pattern-recognition receptors and the activation of the NFκB/RELA pathway to produce innate and homeostatic responses [14,15,16,17]. A prominent feature in the cellular response to RSV replication is the induction of ROS stress, mediated, in part, by the upregulation of cytoplasmic NADPH oxidoreductases producing oxidative modifications including protein carbonylation and DNA damage [18]. The most prominent oxidative DNA modification is the formation of 7,8-dihydro-8-oxo(d)Guanine (8-oxoG; [19]). 8-oxoG is recognized by the base excision repair pathway enzyme, 8-oxoguanine DNA glycosylase1 (OGG1) that initiates a repair complex by hydrolysis of the N-glycosidic bond and cleaving the DNA backbone [20]. Although central to the repair of oxidative DNA damage, OGG1 is now known to be a pleiotropic protein that activates inflammatory genes marked by oxidative DNA damage [21] and also functions as a nuclear-exported stress signal to cooperate in innate inflammation [22,23]. Of note, small-molecule inhibitors of OGG1 reduce inflammation and lung damage in response to TLR3 activation [24]. 

More recently, it has been observed that RSV replication upregulates glucose influx, induces aerobic glycolysis and increases lactic acid generation in epithelial cells. These metabolic adaptations may aid in enhanced nucleoside synthesis and energy needs for viral replication [25,26]. We have extended this work with our finding that RSV infection induces the hexosamine biosynthetic pathway (HBP) [16]. The HBP is a series of enzymatic steps in glucose metabolism, culminating in the formation of UDP-N-Acetyl glucosamine (UDP-GlcNAc), an obligate donor for the glycosyltransferase-catalyzed formation of glycosaminoglycans, glycoproteins and glyco-RNAs [27]. In addition to supporting the production of mature viral glycoproteins, a majority of RSV-induced cellular glycoproteins include extracellular matrix proteins, such as fibronectin and collagen, whose secretion modulate cell–matrix and cell–cell interactions [27]. As a consequence, the HBP is a homeostatic response to RSV that triggers cell-state change and upregulates the cellular capacity for the N-glycosylation of ECM proteins essential in remodeling the basal lamina. 

An important unresolved question is how innate inflammation activated in RSV LRTIs is associated with epithelial plasticity, basement membrane remodeling and long-term changes in lung function. Building on our discovery that the RELA pathway was required for TGFβ induced epithelial plasticity [28], we tested the hypothesis that innate signaling regulates the HBP. We observe that RSV induces intracellular UDP-GlcNAc accumulation and the upregulation of ~171 N glycated peptides within integrins and basal lamina components. To understand this pathway, we observe that the HBP enzymatic pathway is coordinately regulated at the gene transcription level. We observe that RSV induces RELA to interact with regulatory chromatin of pathway genes that is maintained in accessible, nucleosome-free, chromatin determined by ATAC-Seq profiling. Our major findings are that: (1) individually RELA and OGG1 are required for the activation of the HBP pathway genes; (2) RELA and OGG1 are both induced by RSV to bind to the same regulatory domains in the HBP genes; (3) RSV induces RELA∙OGG1 complex formation; and (4) RELA∙OGG1 recruits transcriptionally competent Ser2-phosphorylated RNA polymerase II to the HBP genes. Collectively, these data suggest that the RELA∙OGG1 complex is an epigenetic regulator of the HBP via transcriptional elongation, resulting in UDP-GlcNAC accumulation and N glycation in RSV infection.

## 2. Materials and Methods

### 2.1. Cell Cultures and Treatment

Immortalized primary human small airway epithelial (hSAECs) were grown in SAGM small airway epithelial cell growth medium (Lonza, Walkersville, MD, USA) in 5% CO_2_, conditions that maintain genomic and proteomic signatures representative of primary SAECs [29]. A549 adenocarcinoma cells that maintain type II alveolar cell characteristics (ATCC CCL-185) are grown in minimal essential medium supplemented with 5% fetal bovine serum, streptomycin (100 ug/mL) and penicillin (100 IU). Sucrose-cushion-purified RSV Long strain was prepared as previously described [13,30]. The selective OGG1 inhibitor TH5487 (SelleckChem, Houston, TX, USA) was used at 10 µM and added to the culture medium every 6–12 h. The generation of hSAEC expressing doxycycline (Dox)-inducible RelA shRNA was described earlier [28]. For RELA depletion, experiments, RELA shRNA was induced by addition of Dox (2 µg/mL) to the cell culture medium for 5 days. Control siRNA (D-001810-10-05), RELA siRNA (L-003533-00-0005) and OGG1 siRNA were purchased from Dharmacon (Lafayette, CO, USA). siRNAs were transfected into hSAEC with DharmaFECT1 reagent (Dharmacon). Forty-eight hours later, cells were infected/stimulated. OGG1-deficient cells were generated by CRISPR/Cas9 [23].

### 2.2. Quantification of Cellular UDP-GlcNAc

UDP-GlcNAc was quantified as previously described [31].

### 2.3. Proteomics Analysis of N-glycosylation

Whole cytoplasmic hSAEC extracts were prepared and acetone precipitated by centrifugation 10,000× *g* at 4 °C. The protein pellet was resuspended in 100 μL of 8 M guanidine HCl. About 100 µg of proteins from each sample was digested with trypsin as described previously [32]. The enrichment of N-glycosylated peptides was quantified by the N-glyco-FASP protocol [33]. Peptides were captured by ConA lectin binding and bound to YM-30 Microcon filters (Millipore, Burlingame, MA, USA). Washed filters were incubated with 2 μL N-glycosidase F (Roche, Basel, Switzerland) in 40 μL of 40 mM NH_4_HCO_3_ at 37 °C for 3 h to release deglycosylated peptides. 

Peptides were desalted on a Ziptip C18 (Waters, Milford, MA, USA) and analyzed with a nanoflow Easy nLC1000 UHPLC-Q Exactive Orbitrap mass spectrometer system (Thermo Scientific, San Jose, CA, USA). The peptides were first loaded onto a capillary peptide trap column (Acclaim^®^ Pepmap 100, 75 µm × 2 cm, C18, 3 µm, 100 Å, Thermo Scientific) and then separated on a 25 cm UHPLC reversed-phase column (Acclaim^®^ Pepmap 100, 75 µm × 25 cm, C18, 2 µm, 100 Å, Thermo Scientific) at a flow rate of 300 nL/min. A 2 h linear gradient from 2% solvent A (0.1% formic acid in H_2_O) to 35% solvent B (0.1% formic acid in ACN). Data-dependent acquisition was performed using the Xcalibur 2.3 software in positive ion mode at a spray voltage of 2.1 kV. Survey spectra were acquired in the Orbitrap with a resolution of 70,000, the maximum injection time of 20 ms, an automatic gain control (AGC) of 1e6, and a mass range from 350 to 1600 m/z. The top 15 ions in each survey scan were selected for higher-energy collisional dissociation scans with a resolution of 17,500. Mass spectra were analyzed using MaxQuant software (version 1.5.2.8) [34]. Quantifications were performed with the label-free algorithms in Maxquant [35]. The false positive rate for identification was set to 1% at the peptide level and 1% at the protein level. 

The Maxquant output was analyzed with the Perseus platform [36]. Reversed identifications and common contaminants were excluded from further analysis. After filtering (at least six valid values in total), the remaining missing values were imputed from a normal distribution (width: 0.3 of standard deviation; downshift: 1.8 of standard deviation). Multiple sample ANOVA test with permutation-based FDR correction. The significant hits are those with the q-valve less than 0.01.

### 2.4. RNA Isolation and qRT-PCR

Total cellular RNA was isolated using RNeasy kit with on-column DNase digestion (Qiagen, Germantown, MD, USA). Synthesis of complementary DNAs (cDNAs) was done with a First Strand cDNA Synthesis Kit (ThermoFisher Scientific, San Jose, CA, USA). qRT-PCR assays were performed using SYBR Green Master mix (ThermoFisher Scientific, San Jose, CA, USA) and gene-specific primers (Table 1). Data are presented as fold change using the ΔΔC_t_ method.

### 2.5. Western Blot

Cells were trypsinized, pelleted and washed twice with cold phosphate-buffered saline (PBS). Cell pellets were then lysed in cold low-ionic-strength buffer (50 mM NaCl, 1% IGEPAL, 10% glycerol, 10 mM HEPES, pH 7.4) with fresh added proteinase inhibitor cocktail, DTT and PMSF (ThermoFisher Scientific, San Jose, CA, USA). After the determination of protein concentration, samples were denatured in SDS sample buffer, sonicated three times for 15 s each time (Branson150) and centrifuged at 10,000 rpm for 20 min. Supernatants were heated at 95 °C for 5 min. Proteins were resolved using Criterion TGX 4–15% precast PAGE gels and transferred to nitrocellulose membranes with a Bio-Rad Trans-Blot Turbo transfer system. Primary antibodies used were anti-RelA (ab194726, Abcam, Waltham, MA, USA; MAB5078, R&D, Minneapolis, MN, USA); anti-actin (937215, R&D) was used as a loading control. Blots were imaged and quantified.

### 2.6. Immunofluorescence Microscopy and Flow Cytometry

For immunofluorescence assays, hSAECs were plated on coverslips and infected or treated as indicated. Afterwards, cells were fixed with 4% paraformaldehyde, permeabilized with 0.1% Triton X-100, blocked with 5% goat serum and incubated with primary antibody overnight. Primary antibodies used were anti-GFPT2 (ab190966, Abcam, Waltham, MA, USA). On the second day, coverslips were washed and incubated with Alexa fluor goat secondary antibody. After one hour, cells were washed and mounted using ProLong Diamond Antifade Mountant with 4′,6-diamidino-2-phenylindole (DAPI, Thermo Fisher). The cells were visualized in an ECHO Revolve fluorescence microscope. Individual cell intensities were segmented in FIJI using the StarDist plug-in and plotted as arbitrary fluorescence/cell.

For flow cytometric quantification of GFPT2 expression, hSAECs transfected with control siRNA or RELA siRNA were mock- or RSV-infected (24 h, MOI = 1). Cells were detached with TrypLE, centrifuged and resuspended in 405 live/dead dye (Thermo L34955, ThermoFisher Scientific, San Jose, CA, USA) for 30 min in the dark at room temperature. Cells were washed twice with cold PBS, fixed in 2% paraformaldehyde for 30 min at room temperature and washed 3 times with ice cold PBS. Cells were then permeabilized with 0.1% TritonX-100 in PBS for 20 min at room temperature, blocked with 0.5% BSA and 2% normal bovine serum in PBS on ice for 30 min. After centrifugation, 200 μL of diluted anti- GFPT2 Ab (1:150, Abcam ab190966) was added and incubated on ice for 30 min. The cells were then washed in PBS, 0.5% BSA and 0.05% sodium azide 3 times. A total of 200 μL of a 1:2000 dilution of goat anti rabbit Alex Fluor 642 was added and incubated on ice for 30 min. After three washes, the cell pellet was suspended in 500 μL PBS and assayed by an LSR II (BioRad, Hercules, CA, USA) flow cytometer. Data were analyzed with FlowJo V10.8.0.

### 2.7. Proximity Ligation Assay (PLA)

Anti-RelA (MAB5078, R&D, Minneapolis, MN, USA) and anti-OGG1 (NB100-106, Novus) antibodies (Ab) were used in PLA with Duolink PLA kit (DUO 92010, Millipore Sigma, ThermoFisher Scientific, San Jose, CA, USA) according to the manufacturer’s instructions. The nuclei were counter-stained with DAPI. The PLA signals were visualized in a fluorescence microscope (ECHO) at 20× magnification.

### 2.8. Co-Immunoprecipitation Assay

Interaction between RelA with OGG1 was examined in lysates from mock- or RSV-infected hSAECs (24 h, MOI = 1), prepared in 10mM HEPES pH 7.4, 50 mM NaCl, 1% Triton X-100, 10% glycerol, 1 mM DTT, 1 mM PMSF, and 1× protease inhibitor cocktail (Sigma P8340) and disrupted by sonication (3 times 10 s, at 4 °C). Cleared lysates were incubated with either anti-OGG1 (Novus Biologicals NB100-106, Centennial, CO, USA) or anti-RELA Ab (R&D, MAB5078, Minneapolis, MN, USA) overnight at 4 °C, then captured on protein G Dynabeads (Invitrogen 10-004-D, ThermoFisher Scientific, San Jose, CA, USA), eluted in SDS-PAGE buffer followed by fractionation by 4–15% TGX gel (BioRad) and transferred to PVDF membrane and immunoblotting with either anti-RELA or anti-OGG1 Abs as indicated. For immune complex analysis of FLAG-tagged OGG1, lysates from mock- or RSV-infected cells were captured on anti-FLAG M2 Magnetic Beads (Cat # M8823, Millipore Sigma, ThermoFisher Scientific, San Jose, CA, USA) [37]. Samples were divided into input (to determine levels of total RELA) or incubated with preequilibrated anti-FLAG magnetic beads. After binding, beads were then washed; complexes were eluted and resolved on 4–15% PAGE gels (BioRad, Hercules, CA, USA) as described above and subjected to Western immunoblot analysis using phospho-RELA Ab (ser 256; Santa Cruz Biotechnology sc-136548, Santa Cruz, CA, USA). The membrane was stripped and blotted with mouse monoclonal anti-FLAG (Millipore Sigma F1804) antibodies. Total RelA was determined using anti-NFκB RelA Ab (Santa Cruz Biotechnology sc-8008, Santa Cruz, CA, USA).

### 2.9. Two-Step Chromatin IP (XChIP)-Quantitative Genomic PCR (Q-gPCR)

Protein–protein cross-linking was performed with DSG (2 mM, 45 min at 22 °C) followed by protein–DNA cross-linking with formaldehyde [38]. Equal amounts of sheared chromatin were immunoprecipitated (IPed) overnight at 4 °C. IPs were collected with 40 μL protein-A magnetic beads (Dynabeads, ThermoFisher Scientific, San Jose, CA, USA), washed and eluted in 250 µL elution buffer for 15 min at 65 °C. Gene enrichment was determined by Q-gPCR using region-specific PCR primers (Table 2). The fold change of DNA in each IP was determined by normalizing the absolute amount to input DNA reference and calculating the fold change relative to that amount in unstimulated cells [39].

### 2.10. Statistical Analyses 

Statistical analyses were performed with Graph Pad Prism 9 (GraphPad Software, San Diego, CA, USA). Results are as individual replicates in box plots as median ±10–90%. For multiple group experiments, one-way ANOVA was used with post-hoc Tukey T-tests for a group-wise comparison between treatments. *p* values < 0.05 were considered to be statistically significant.

## 3. Results

### 3.1. RSV Induces the HBP in an NFκB/RELA-Dependent Manner

In this study, we focus on hSAECs because these maintain representative genomic and proteomic signatures of small airway epithelial cells [29] and, in response to RSV replication [14], produce the same spectrum of pathogenic disease-modifying cytokines [29] and undergo inducible cell-state changes [28,39,40] without exhibiting senescence. hSAECs were infected with sucrose-cushion-purified RSV, and the HBP pathway was examined. The HBP pathway is composed of five enzymes responsible for four metabolic steps in the conversion of fructose 6 phosphate and glutamine into UDP-GlcNAc (Figure 1A). 

To determine whether RELA regulates the HBP, hSAECs expressing a doxycycline (DOX)-inducible RELA shRNA was first used. This approach enables the study of the effect of RELA depletion without the well-described compensation for NFκB deficiency in knockout cell lines [40]. We first confirmed the effect of Dox treatment on RELA abundance, wherein 5 days of Dox treatment reduced endogenous 65 kDa RELA to <15% that of untreated hSAECs (Figure 1B). At the gene expression level, we found that RELA mRNA expression is induced 2.8-fold after RSV infection; both basal and RSV-induced RELA mRNA expression was reduced to that of the control in the DOX-treated cells, indicating a robust depletion (*p* = 0.029, *n* = 3, Figure 1C). To confirm that this level of RELA depletion was sufficient to block the expression of NFκB-dependent genes, we observed that the 39-fold induction of IL6 mRNA by RSV in wild-type cells was decreased to 12-fold in the RELA-depleted cells after DOX treatment (*p* < 0.0001, *n* = 3; Figure 1D). Collectively, these data indicated the functional depletion of RELA signaling in response to DOX treatment.

We next asked whether the HBP enzymatic pathway and metabolic output was regulated by RSV in an RELA-dependent manner. Using the shRNA depletion model, the cytoplasmic abundance of UDP-GlcNAc was measured using a quantitative LC-MS assay developed earlier by us [31]. We observed that RSV induced a ~2-fold increase in cellular UDP-GlcNAc in wild-type cells (*p* = 0.016, *n* = 3, Figure 1D). By contrast, there was no significant induction of UDP-GlcNAc in the RELA shRNA knockdowns (Figure 1D). 

To prove that this DOX-dependent phenomenon was not an artifactual result of clonal selection, we confirmed the RELA-dependence of the HBP pathway in wild-type hSAECs after siRNA-mediated transfection. In comparison to scrambled (Scr)-transfected cells, the population of RELA siRNA-transfectants had significantly lower basal and RSV-induced RELA mRNA expression (*p* < 0.0001, Figure 1E). Similarly, the RELA depletion was functionally significant, inhibiting the 42-fold increase of IL6 expression to 19.9 fold (*p* < 0.0001, *n* = 3, Figure 1E). Likewise, the ~2-fold accumulation cytoplasmic UDP-GlcNAc by RSV infection was reduced by RELA depletion and was not different from control, mock-infected wild-type cells (Figure 1F). 

### 3.2. RELA Is Required for RSV-Induced N Glycoproteome Production

The accumulation of UDP-GlcNAc, an activated donor for protein N glycosylation, suggested to us that RSV increases the abundance of N glycoproteins in vivo in a RELA-dependent manner. N glycosylation is required for the folding, functional activity and secretion of major ECM proteins upregulated by epithelial plasticity, notably fibronectin and collagen [16,31]. However, the identities of these glycoproteins are not known. To advance this understanding, we profiled the cellular N glycoproteome of hSAECs in response to RSV infection in the absence or presence of RELA by siRNA knockdown. Glycoproteins were enriched by lectin affinity and filter-immobilized. N-linked glycoproteins were released from by N-glycosidase F, N-glyco-FASP and identified by LC-MS/MS (Methods, Ref [41]).

We observed a robust increase of N-glycosylation of 171 peptides by RSV infection (Figure 2A). Remarkably, this induction was blocked after RELA depletion (Figure 2A). To examine the peptides induced by RSV infection, the peptides were analyzed by a Volcano plot, a visualization that displays the fold change in the peptide abundance (RSV versus mock) versus log2 (ANOVA) of its significance of identification (Figure 2B). A functional analysis of the pathways enriched in the proteins within this group identified the statistical over-enrichment of integrin (ITG) signaling pathways (Panther pathway P00034, FDR = 4.67E-07). Within this pathway, multiple peptides of six ITG proteins were identified, which included ITG-A1, -A3, -A5 and -B6 isoforms (Figure 2B). A similar Volcano plot comparison was conducted to identify those proteins that were RELA-dependent, comparing peptides in RSV-infected wild-type versus RSV-infected RELA-deficient cells (Figure 2C). This analysis also showed the enrichment of the ITG signaling pathway. The mean signal intensities of the six most highly upregulated and significant peptides in the ITG signaling pathway are shown in Figure 2D, wherein ITG-A3, -A5, -A6, galectin-3 binding protein (LGALS3BP), ICAM1 and laminin subunit gamma 1 (LAMC1) are shown. An overlapping set of N-linked glycoproteins were identified after RELA silencing by shRNA (Appendix A). Together these data indicate that RSV induces the glycosylation of integrin- and basement membrane proteins in an RELA-dependent manner.

We therefore sought to investigate the mechanism for HBP activation at the molecular level. 

### 3.3. RELA Coordinately Regulates Core Enzymes of the HBP

We first examined the effect of RSV on the expression of the five HBP core enzymes controlling the four metabolic steps in UDP-GlcNAc synthesis and their dependence on RELA signaling. qRT-PCR was performed on hSAEC transfected with Scr-siRNA or RELA-directed mRNA and infected in the absence (mock) or presence of RSV. We observed that glutamine--fructose-6-phosphate transaminase 1 (*GFPT1)* mRNA was induced 1.5-fold by RSV (*p* = 0.001, *n* = 4, Figure 3A); this induction was reduced to control levels by RELA siRNA transfection (Figure 3A). We noted that *GFPT2* was the mostly highly upregulated mRNA of the HBP pathway, with basal GFPT2 being barely detectable but steady-state levels increasing by ~37-fold in response to RSV infection over mock-infected cells (Figure 3B). This RSV induction of *GFPT2* mRNA was significantly decreased in RELA-depleted cells (36.2-fold to 21.2-fold, *p* < 0.0001, *n* = 4, Figure 3B). Similar patterns of RSV inducibility and reversal by RELA depletion was seen for the 4.1-fold induction of Phosphoglucomutase 3 (*PGM3)* mRNA (Figure 3C), the 7.7-fold induction of UDP-N-Acetylglucosamine pyrophosphorylase 1 (*UAP1)* mRNA (Figure 3D) and the 4-fold induction of *GNPNAT1* mRNA (Figure 3E). To exclude the possibility that RSV replication was affected by RELA depletion, the expression of RSV N protein was measured; there was no significant difference between the Scr and RELA transfectants (Figure 3F). From this experiment, we concluded that RELA coordinately activates the major components of the HBP by inducing gene expression.

To confirm that the changes in mRNA reflect those of the protein abundance, we validated the expression of GFPT2 by semi-quantitative confocal immunofluorescence. Control or siRNA-transfected hSAECs were mock- or RSV-infected and stained with anti-GFPT2 Ab. In mock-infected cells, we observed faint cytoplasmic GFPT2 staining that was enhanced in response to RSV infection in a heterogeneous manner, with some cells staining intensely (Figure 4). By contrast, in siRNA-transfected cells, the number and overall intensity of GFPT2 staining was reduced but not eliminated.

To measure the population response, fluorescence intensity was segmented for each cell and quantified as fluorescence intensity/cell (Figure 4B). To confirm changes more quantitatively, hSAECs were analyzed by flow cytometry (Figure 4C; gating strategy is shown in Appendix A). These data indicate that 1. RSV significantly increases cytoplasmic GFPT2 expression; 2. basal GFPT2 expression is controlled by RELA; and 3. RSV-inducible GFPT2 expression is partly activated by RELA. These data are consistent with an RELA-independent mechanism for the activation of GFPT2 expression. To this end, we note that GFPT2 is independently regulated by the unfolded protein response through XBP1s [16] and propose that this pathway partly compensates for RELA in the siRNA-transfectants. 

### 3.4. RSV Induces RELA Binding to Open Chromatin in HBP Genes

To explore the molecular mechanism for HBP pathway upregulation, we examined the effect of RSV on RELA binding to open chromatin domains previously identified by a Assay for Transposase-Accessible Chromatin (ATAC-Seq) study. These domains overlap with known H3K27 acetylated binding domains, signatures characteristic of active enhancer elements. A large open chromatin domain on the most highly inducible HBP pathway gene, *GFPT2*, was found within the gene itself (Figure 5A, left panel). Other open chromatin domains were identified overlapping with the proximal promoters for Glucosamine-Phosphate N-Acetyltransferase 1 (*GNPNAT1), PGM3* and *UAP1* (Figure 5B–D, left panels).

To determine whether RSV induced RELA binding to these open chromatin domains, we applied a two-step chromatin immunoprecipitation technique developed by us and optimized for detecting RELA binding [39]. We observed that RSV induced a 35-fold increase of RELA binding to the *GFPT2* intragenic enhancer, a 16-fold induction to *GNPNAT1* promoter, a 220-fold increase to the *PGM3* promoter and a 40-fold induction to the *UAP1* promoter (Figure 5B–D, right panels; note that these findings were replicated in three separate experiments using two independent replicates, each measured by technical duplicates).

### 3.5. HBP Pathway Genes Are OGG1-Dependent

Earlier work has shown that OGG1 recognizes oxidative-stress-induced 8-oxoG lesion as an epigenetic-like mark in regulatory promoters [42]. OGG1 binding to 8-oxoG in regulatory elements promotes NFκB-dependent inflammatory gene expression including *TNF, CLL20, IL6* and *CCL5* associated with RSV-induced pulmonary neutrophilia in vivo [23]. The role of OGG1 in the induction of RSV-induced epithelial plasticity and HBP have been unexplored. To determine whether the expression of HBP pathway enzymes were OGG1-dependent, hSAECs were transfected with scrambled (Scr)- or OGG1-targeting siRNA and infected with RSV. We observed that the RSV-induced expression of *GFPT2* mRNA was substantially reduced by OGG1 depletion. Herein, RSV produced a 27.8-fold induction of *GFPT2* mRNA in Scr-RNA transfected hSAECs, and expression was reduced to 16-fold in OGG1-silenced hSAECs (*p* < 0.0001, *n* = 4, Figure 6A). To extend this data, we asked whether OGG1 activity was required for GFPT2 expression. For this purpose, cells were treated in the absence or presence of TH5487, a selective small-molecule OGG1 active site that blocks the binding and repair of 8-oxoG [24]. We observed that TH5487 treatment blocks *GFPT2* mRNA expression in hSAECs (from 17.5-fold to 6-fold, *p* < 0.0001, *n* = 4, Figure 6B). This OGG1 dependence was also seen in transformed A549 lower epithelial cells, wherein the 8-fold *GFPT2* increase was reduced to control values (*p* < 0.0001, *n* = 4. Figure 6C). These results were confirmed in hSAECs depleted of OGG1 by CrispR/Cas9 [23], wherein a 7.3-fold induction of *GFPT2* mRNA expression was completely silenced in the OGG1^−/−^ genotype (*p* < 0.0001, *n* = 3, Figure 6D). The OGG1 dependence of *UAP1* and *GNPNAT1* was also confirmed in TH5487-treated hSAECs (Figure 6E,F). These data indicate that both RELA and OGG1 were required for the expression of the HBP core pathway genes.

To confirm TH5487 affected the expression of GFPT2, semi-quantitative fluorescence microscopy was performed. We observed the heterogeneous expression of GFPT2 expression by RSV was substantially reduced by the TH5487 treatment (3.4 AU in solvent-treated cells to 2.6 AU in TH5487-treated cells, *p* = 0.006, Figure 7). These data indicate OGG1 is also required for the expression of the HBP core enzymes previously noted to be RELA-dependent.

### 3.6. RSV Induces RELA∙OGG1 Complex Formation

Based on our earlier finding that OGG1 forms a complex with RELA on promoters of inflammatory genes [23], we next asked whether RELA and OGG1 formed an RSV-inducible complex. For this purpose, we conducted proximity ligation assays (PLAs). PLAs detect atomic-distance interactions in situ detected by the enzymatic ligation of separately directed antibody-conjugated oligonucleotides that are amplified by PCR, appearing as red foci in immunofluorescence. In the absence of RSV infection, a few RELA∙OGG1 complexes are detected, distributed both in the nucleus and cytoplasm. The cytoplasmic distribution appears in an organellar pattern (OGG1 is known to localize to the inner mitochondrial membrane [43]). By contrast, in response to RSV infection, numerous foci are produced, especially abundant in nascent syncytia (Figure 8). 

To independently validate the protein–protein interaction between OGG1 and RELA, coimmunoprecipitation experiments were performed. hSAECs were mock- or RSV-infected, and total cellular lysates were immunoprecipitated (IPed) with anti-OGG1 antibody. The complexes were then captured on protein G-conjugated Dyna beads and fractionated for immunoblot (IB) with anti-RelA. In mock-infected cells, a faint staining of 65 kDa RELA was observed that was dramatically increased by RSV infection (Figure 9A). To confirm the OGG1∙RELA association, mock- or RSV-infected cells were IPed with anti-RELA Ab, and the immune complexes were probed for OGG1 by Western. In mock-infected cells, a faint 50 kDa OGG1 band was observed that was increased by RSV infection (Figure 9B).

To further explore this interaction, we asked whether RSV induced phosphorylated RELA to bind the OGG1 complex. In this experiment, FLAG-OGG1-expressing cells were infected with a time course of RSV infection and the lysates subjected to IP with anti-FLAG M2 antibody-conjugated beads. The immune complexes were then subjected to Western blotting with anti-phospho Ser536 RELA antibody. We observed a time-dependent increase in 65 kDa phospho-Ser536 RELA recruitment into the complex (Figure 9C). Collectively, these data validate that RSV induces RELA∙OGG1 complex formation.

### 3.7. OGG1 Binds to the GFPT2 Intragenic Enhancer

To further understand the dependence of the HBP pathway genes on OGG1, we applied the XChIP to see if OGG1 interacted with the open chromatin domains. Wild-type hSAECs were mock- or RSV-infected and subjected to XChIP using anti-OGG1 Ab. We observed that OGG1 showed substantial binding to the *GFPT2* intragenic enhancer in uninfected cells, with a 40-fold increase in binding relative to input IgG control (Figure 10A). Upon RSV infection, an additional increase in OGG1 binding was observed (110-fold; *p* < 0.01 compared to uninfected cells, Figure 10A). A similar pattern of inducible OGG1 binding was observed for *UAS1* (Figure 10B).

### 3.8. RSV-Induced Recruitment of RELA Is OGG1-Dependent

We next asked whether OGG1 binding was coupled to RELA recruitment. Cells were preincubated with solvent or OGG1 inhibitor and mock- or RSV-infected prior to XChIP examining the binding of RELA. RSV induced a ~2-fold induction of RELA binding to the intergenic *GFPT2* enhancer, which was inhibited by TH5487 (Figure 10C). A similar pattern of RSV induction and inhibition by the OGG1 inhibitor was observed for the UAS1 promoter (Figure 10D).

### 3.9. OGG1 Is Required for Recruitment of Transcriptional Elongation-Competent Pol II

We next asked the question whether OGG1 was required for the transcriptional elongation of the HBP genes. For this purpose, we examined the effect of RSV and OGG1 inhibitor on the binding of phospho-Ser 2 carboxyterminal domain (CTD) Pol II, the transcriptional elongation-competent form of Pol II. We found that RSV infection induced a substantial increase in phospho-Ser2 CTD RNA Pol II binding (1236-fold vs 515-fold over IgG control, *p* = 0.0018, *n* = 3, Figure 10E). The inhibition of OGG1 substantially reduced the phospho-Pol II binding to control levels (*p* = 0.005, *n* = 3, Figure 10E). TH5487 significantly inhibited the recruitment of phospho-Ser2 CTD RNA Pol II to the *UAS1* promoter (Figure 10F, *p* = 0.0028, *n* = 3).

## 4. Discussion

RSV is a major human pathogen for which no effective vaccine or therapy has yet been developed. In particular, severe LRTIs produced by RSV are a major cause of pediatric hospitalizations and adult morbidity. Not only are LRTIs in infants associated with long-term decreased pulmonary function, repetitive infections in adults causing exacerbations of asthma are associated with declines in FEV1 [5,6,9,10]. Studies of fatal pediatric infections, human challenge studies and preclinical rodent models have all demonstrated that RSV replicates in the small bronchiolar airway epithelial cells [4,12], activating a cellular innate response. RSV replication in the lower airway is important for disease pathogenesis, wherein replication in SAECs is associated with epithelial sloughing, leukocytosis and bronchiolar obstruction [4,44]. Although a number of diverse epithelial cell types are present in the small bronchioles, a population of epithelial cells derived from Scgb1a1-/CC10-expressing progenitors is thought to play an important role. These cells undergo virus-induced plasticity, exhibit oxidative stress and express a predominance of neutrophil-recruiting and Th2-polarizing cytokines [12,13,29]. Upon entry, RSV activates recognition receptors upstream of the RELA pathway to produce innate anti-viral and anti-oxidant responses, triggering cellular plasticity important in homeostasis and injury repair [14,15,16,45]. Our study herein addresses an incompletely resolved mechanism of how innate inflammation cooperates with oxidative stress to trigger metabolic reprogramming and basement membrane remodeling.

The major findings of this study elucidate a dynamic interplay between RSV induced innate signaling, DNA damage repair and metabolic reprogramming of hexosamine biosynthesis supporting integrin signaling. Our novel finding of the RSV-induced complex formation of activated RELA and OGG1 advances understanding how RSV induces the remodeling of enhancers; the recruitment of processive RNA polymerase to transcribe key HBP components, producing intracellular UDP-GlcNAc accumulation; and the N glycosylation of cytoplasmic integrins and basement membrane proteins.

### 4.1. The Hexosamine Biosynthetic Pathway (HBP) 

The HBP is responsible for diverting 2–5% of intracellular glucose into UDP-GlcNAc, an energy donor important in protein folding, ECM secretion and regulatory post-translational modifications of proteins. The role of HBP has been extensively described in metastatic transformation as a pathway responsible for diverting fructose 6-phosphate into pathways that produce nucleosides, amino acids, macromolecules and organelles required for invasiveness and proliferation [46]. The role of the HBP in homeostasis and metabolic adaptation of primary epithelial cells in response to viral infections is not fully known. Our studies extend the role of inducible HBP in virus-induced epithelial plasticity, integrin signaling and ECM remodeling driven by N-linked glycosylation.

Earlier, we made the discovery that TGFβ-induced epithelial plasticity activated the HBP to relieve cells from proteotoxic stress produced by the enhanced accumulation of ECM proteins, including fibronectin and collagen, in the ER. Here, the activation of the HBP enhances the N-glycosylation of misfolded insoluble ECM components, permitting their folding, processing N-glycan branching and secretion [47]. Through enhancing protein folding, the upregulation of the HBP prevents proteotoxicity from the enhanced production of ECM-modifying proteins that trigger the unfolded protein response [31].

Our studies clearly demonstrate that the HBP is highly inducible by RSV infection in primary epithelial cells. Our earlier work showed that the HBP is independently induced by the IRE1-XBP1 signaling arm of the unfolded protein response [16]. XBP1 is a transcription factor that activates a distinct subset of the HBP genes, including GFPT2. We extend understanding of HBP pathway regulation to include the RELA∙OGG1 complex as one that independently regulates *GFPT2, PGM3, UAP1* and *GNPNAT1*. Earlier we showed that IRE1-XPB1 also regulates GFPT2 of the HBP pathway important in UDP-GlcNAc accumulation, suggesting that multiple stress-activated cellular signaling pathways control the HBP [16]. Interestingly, our study indicates that GFPT2 is more highly inducible by RELA than the GFPT1 isoform. GFPT2 has been shown to be less sensitive to UDP-GlcNAc feedback inhibition [48], which may account for the higher intracellular UDP-GlcNAc levels in RSV infection. We are intrigued by a separate finding that XBP1 directly associates with the rate-limiting kinase of RELA activation, IκB kinase [32], in response to TGFβ stimulation. Whether IKK∙XBP1 complex is formed in response to RSV infection, and how this pathway interfaces with HBP activation will require further study.

### 4.2. RELA-Dependent N-glycosylation of Integrin Signaling Pathway

N-glycosylation refers to the enzymatic attachment of a carbohydrate to the amide nitrogen of asparagine and is one of the most abundant post-translational modifications of protein [47], and it is known to play important roles in cell adhesion and migration by modulating the conformation and function of cellular adhesion molecules. The aberrant N-glycosylation of integrins has been observed in innate inflammation, cancer progression and metastasis [49]. Over 100 glycosyltransferases have been identified and are known to be regulated by changes in cellular microenvironment and cellular metabolism [50], our work here identifies the activated RELA∙OGG1 complex in controlling the inducible N-linked glycosylation of a small component of the cellular proteome enriched in integrin signaling, basement membrane composition and ECM. Although some of the functional effects of N glycosylated sites are unknown and beyond the scope of this manuscript, we speculate that N glycosylation of integrins may affect cellular motility and/or recognition of ECM. Additionally, N glycosylated ECM components may also affect integrin signaling. The role of N glycosylated ICAM in epithelial motility in virus induced inflammation and/or plasticity will require further investigation.

### 4.3. RELA Is a Master Transcription Factor of Epithelial Plasticity Linking Innate Inflammation to Expression of EMT Regulators

RELA is a major mediator of the innate immune response, controlling a global network of virus-inducible interferons (IFNs) and IFN-stimulated genes (ISGs) by the recruitment of a chromatin remodeling protein, BRD4, that couples RELA binding with transcriptional elongation [17]. Using next-generation sequencing of RELA-silenced cells, we found that RELA is also the master regulator of TGFβ-induced epithelial plasticity, wherein it is responsible for expressing a cascade of master transcription factors AP1, SNA1 and ZEB that maintain epithelial–mesenchymal transition [28].

In this study, we extend further the role of RELA as an effector RSV-induced HBP gene expression, UDP-GlcNAc accumulation and the N-linked glycosylation of 171 peptides. Chromatin immunoprecipitation experiments indicate that RELA interacts with the regulatory domains of the HBP. These domains are enriched in H3K27Ac, an epigenetic mark characteristic of active enhancers [51]. We propose that the RELA∙OGG1 complex accesses and remodels the accessible/nucleosomal-free HBP regulatory domains to coordinately express the HBP enzymes through a process of transcriptional elongation. Transcriptional elongation is a highly regulated gene expression mechanism widely employed in the innate immune response. In this process, RELA binding triggers the recruitment and transition of processive RNA Pol II to genes maintained in an open chromatin conformation [52,53,54]. These studies extend the mechanism of OGG1 in gene expression control by demonstrating OGG1 is required for initiating transcriptional elongation.

Our immunofluorescence data indicate that there is substantial heterogeneity in the RELA-dependent expression of GFPT2. This is an expected result based on previous analysis of virus-induced innate signaling using single-cell imaging and computational modeling. These studies found that the RELA pathway shows substantial cellular variability, e.g., stochastic behavior, due to the differences in cellular geometry, initial RELA concentrations and the translational rate of the IκB inhibitors [55,56]. We therefore expect the downstream RELA-dependent genes to exhibit this wide dynamic range, perhaps explaining the results in immunofluorescence. A corollary to this observation is that some epithelial cells will be able to mount a successful response to viral induced proteotoxicity, whereas others will be destined for apoptosis.

### 4.4. OGG1-In Base Excision Repair and as an Epigenetic Regulator of HBP

OGG1 constitutes a frontline defense of oxidative DNA damage by repairing 8-oxoG in the base excision repair pathway [57]. Unrepaired oxidative DNA base lesions are not only mutagenic but also contribute to disease pathogenesis in RSV infection [18,19,58]. Recent studies identified that, in guanine-rich gene regulatory regions, 8-oxoG functions as an epigenetic-like mark, and, in stress conditions, its binding by OGG1 has distinct roles in the modulation of gene expression [42]. The role of OGG1 in facilitating pro-inflammatory gene expression is best understood in tumor necrosis factor-α stimulation, wherein OGG1-deficient cells accumulate oxidized DNA and show diminished *CCL20, IL6, CCL5* and *CXCL10* expression. Studies complementing OGG1 deficient cells with an OGG1 mutant defective in BER (K249Q) showed that the enzymatic activities of OGG1 were dispensable for gene expression. Under high amounts of oxidative stress, OGG1 itself is oxidized, inactivating its enzymatic activity. The oxidatively disabled OGG1, stalled at 8-oxoG binding, nucleates the binding of transcription factors [37,59,60].

The development of TH5487 as a selective small-molecule OGG1 inhibitor has provided a useful probe for evaluating the epigenetic role of OGG1. In cocrystal, TH5487 binds in the hydrophobic oxoG binding pocket of OGG1, inducing acute pharmacological deficiency and preventing OGG1-DNA interactions at the guanine-rich promoters of proinflammatory genes [24]. More recently, we have found that OGG1 facilitates RELA binding to 8-oxoG modified regulatory elements in cytokine promoters [37], a phenomenon reproduced in this study for the HBP pathway genes. These studies, however, have not examined the role of OGG1 in epithelial plasticity, transcriptional elongation or N-glycosylation. Our findings that OGG1 inhibitor blocks RSV-induced phospho-Pol II recruitment provides an important mechanistic link for how OGG1 binding to epigenetic regulatory regions facilitates gene expression.

Our protein association studies, including PLA and Co-IP indicate that the RELA∙OGG1 complex is highly induced by RSV infection. The PLA studies indicate that the complex is primarily in the nuclei of infected cells. Despite a prominent role for the RELA∙OGG1 complex in nuclear gene expression, some of the complex is found in perinuclear regions in the cytoplasm. The most likely explanation for this phenomenon is that RSV infection, a potent inducer of mitochondrial oxidative stress, results in import of OGG1 into the mitochondria. OGG1 contains a COOH terminal mitochondrial localization sequence and has been shown to localize to mitochondria during oxidative stress, wherein it participates in mitochondrial DNA repair [43]. Similarly, RELA also has been observed to translocate into the mitochondrion [61]. It will be of interest to determine whether the RELA∙OGG1 complex is involved in the mitochondrial gene expression required for oxidative phosphorylation during RSV infection.

Our Co-IP studies confirm the presence of the RSV-inducible RELA∙OGG1 complex and extend this understanding to show that the complex contains Ser 536 phosphorylated RELA. Ser536 phosphorylation is thought to be an IκBα-independent activation pathway for NFκB [62]; our earlier studies have demonstrated that Ser536 is involved in the activation of inflammatory genes in response to oxidative-stress-inducing ligands [63]. Whether other phosphorylated forms of RELA are associated with the OGG1 complex will require further investigation. 

## 5. Conclusions

We conclude that RSV activates the small airway epithelial HBP through the transcriptional activation of four primary metabolic enzymes responsible for intracellular UDP-GlcNAc accumulation (Figure 11). UDP-GlcNAc is an obligate donor of N-linked glycoprotein formation enhances the N glycation of integrin and basal lamina proteins.

We implicate the RSV-inducible RELA-OGG1 complex in interacting with active regulatory elements within accessible chromatin domains recruiting activation RNA polymerase and gene expression. We find that OGG1 interacting with nucleosomal-free chromatin regulatory elements in the HBP gene is required for the recruitment of activated transcriptional elongation complexes important in regulated transcription. We conclude that the RELA∙OGG1 complex is an epigenetic regulator mediating metabolic reprogramming and the N glycoprotein modifications of integrins in response to RSV. These findings have implications for RSV induced adaptive epithelial responses and extracellular matrix remodeling that may be important in long-term changes in lung function in indviduals with severe or repetitive RSV infections. 

## Figures and Tables

**Figure 1 cells-11-02210-f001:**
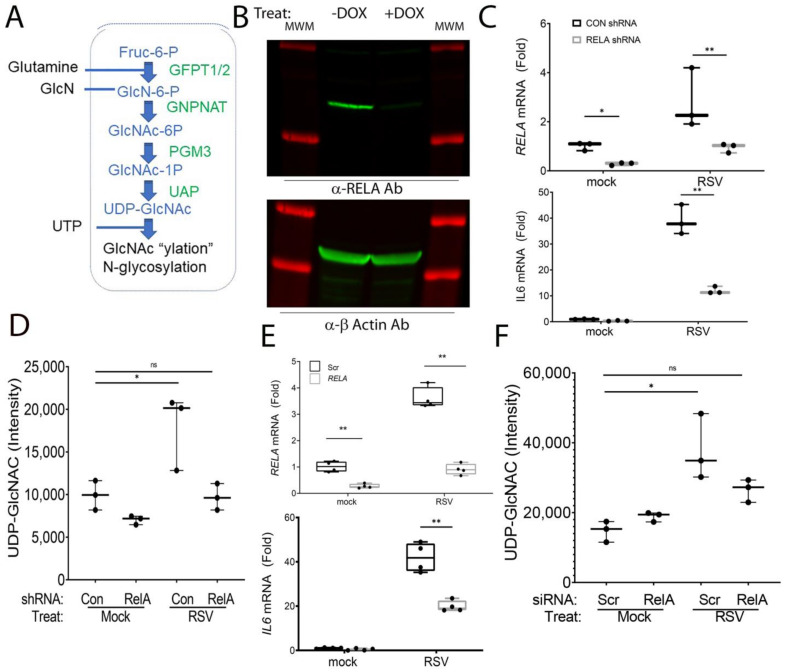
RELA regulates UDP-GlcNAc synthesis. (**A**) Schematic view of the HBP pathway from Fructose-6 phosphate (Fruc-6P) to the formation of UDP-GlcNAc. Metabolites are colored in blue; enyzmes are green. UDP-GlcNAc is a donor for the N-glycosylation of extracellular matrix (ECM) proteins, as well as protein GlcNAc-ylation. Abbreviations are: GFPT, Glutamine-Fructose-6-Phosphate Transaminase; PGM, phosphoglucomutase; UAP, UDP-N-Acetylglucosamine Pyrophosphorylase; and GNPNAT, Glucosamine-Phosphate N-Acetyltransferase. (**B**) Western blot of RELA in doxycycline-regulated hSAECs. Cells were cultured in the absence (-) or presence (+) of DOX before whole cell lysis. Top panel, anti-RELA staining (green); bottom panel, β-actin staining (green) as housekeeping control. The prominent 65kDa RELA staining is depleted after DOX treatment. MWM, molecular-weight markers (red). (**C**) qRT-PCR. Cells cultured in the absence or presence of DOX were mock-infected or RSV-infected for 24 h prior to qRT-PCR. Data are expressed as a fold change of the untreated control, normalized to PP1A. Each symbol is a biological replicate (*n* = 3). Top panel, RELA mRNA; bottom panel, IL6 mRNA. Data are significant by ANOVA. **, *p* < 0.01 by post-hoc Tukey comparison. (**D**) Cytoplasmic UDP-GlcNAc abundance. Cells were treated as in panel (**C**), and total cellular UDP-GlcNAc was measured by LC-MS/MS. *, *p* < 0.01. (**E**) qRT-PCR of RELA depleted hSAECs by siRNA. hSAECs were transfected with scrambled (Scr) or RELA-targeting siRNA. Top panel, fold change of RELA mRNA; bottom panel, IL6 mRNA. (**F**) UDPG-GlcNAc in siRNA- transfected cells. Intracellular UDP-GlcNAc was measured in control siRNA or RELA-siRNA transfected cells in the absence or presence of RSV infection. *, *p* < 0.01; ns, not significant.

**Figure 2 cells-11-02210-f002:**
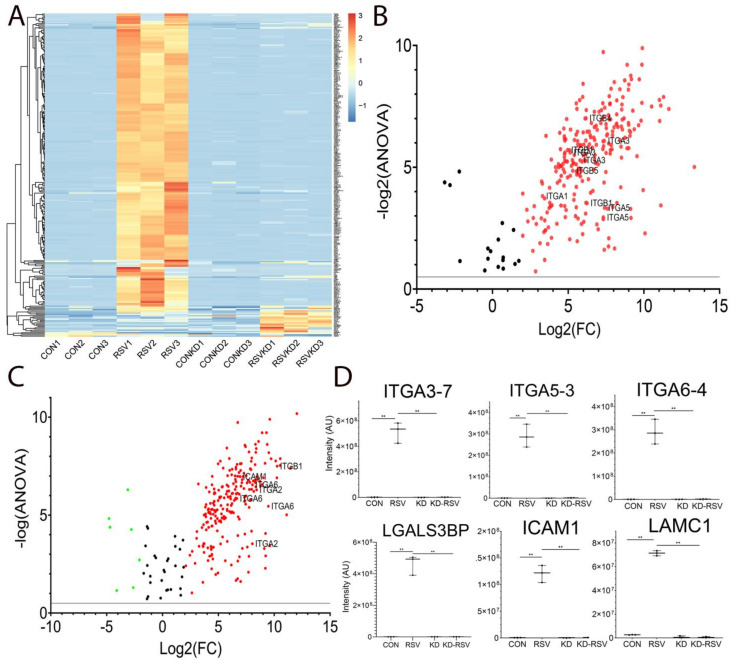
RSV and RELA regulate intracellular N-glycosylation of the integrin signaling pathway. N-glycoproteins were identified in cytoplasmic proteins from control and RSV-infected cells in the presence or absence of RELA silencing. (**A**) Agglomerative hierarchical clustering of 171 peptides significantly changed across the treatment conditions. Data was Z-scored across the row prior to clustering using Euclidian distance. The legend is at right. Each row represents an individual peptide; columns refer to different cell treatments/conditions. Note that the replicates for each treatment group cluster together and that the bulk of the N-glycopeptides are induced by RSV infection and are not observed in the RELA-depleted cells. (**B**) Volcano plot of peptides in RSV- versus mock-treated samples. X axis, fold change RSV versus mock; Y axis, log2 (ANOVA). Selected peptides in the integrin signaling pathway are labeled. (**C**) Volcano plot of peptides in RSV-infected cells WT versus RELA knockdown samples. X axis, fold change RSV versus mock; Y axis, log2 (ANOVA). Selected peptides in the integrin signaling pathway are labeled. (**D**) LFQ intensity for selected integrin (ITG), galectin-3 binding protein (LGALS3BP), ICAM1 and laminin subunit gamma 1 (LAMC1) peptides. **, *p* < 0.01 post-hoc.

**Figure 3 cells-11-02210-f003:**
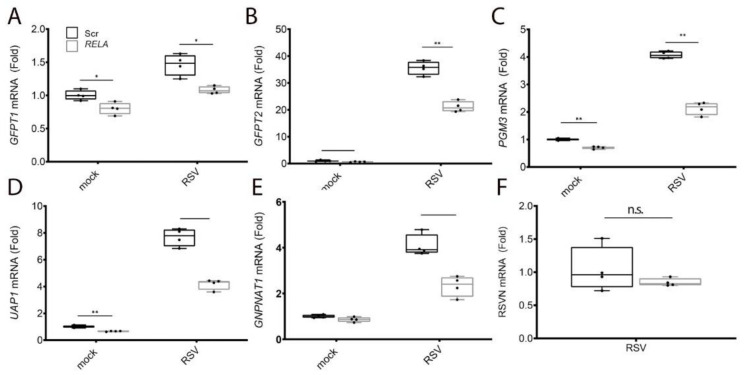
RELA coordinately activates expression of the HBP. Scrambled (Scr) or RELA-directed siRNA was transfected into hSAECs. Cells were mock-infected or RSV-infected (MOI = 1, 24 h) and qRT-PCR performed. Data are presented as fold change untreated controls normalized to peptidylprolyl isomerase A (PPIA) (**A**) GFPT1 mRNA, (**B**) GFPT2 mRNA, (**C**) PGM3 mRNA, (**D**) UAP1 mRNA, (**E**) GNPNAT1 mRNA. **, *p* < 0.01; *, *p* < 0.05, by post-hoc Tukey comparison. (**F**) Expression of RSV N transcript. n.s., not significant.

**Figure 4 cells-11-02210-f004:**
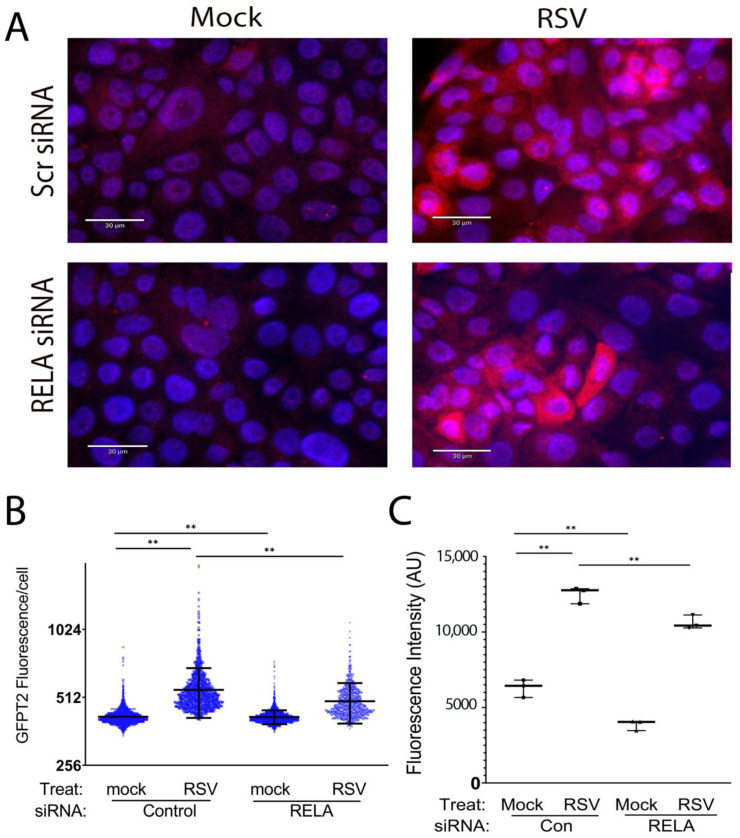
Role of RELA in cytoplasmic GFPT2 expression. (**A**) Scr or RELA siRNA-transfected hSAECs were mock-infected or RSV-infected. Cells were permeabilized and stained with anti-GFPT2 Ab (red). Nuclear stained is by DAPI (blue). Bar indicates 30 μm. (**B**) The quantitation of GFPT2 staining by FIJI. GFPT2 fluorescence was segmented by StarDist and plotted as arbitrary fluorescence/cell for each treatment condition. **, *p* < 0.01 by post-hoc Tukey comparison. (**C**) Flow cytometric quantification of GFPT2. GFPT2 fluorescence intensity was measured by flow cytometry. **, *p* < 0.01 by post-hoc Tukey comparison.

**Figure 5 cells-11-02210-f005:**
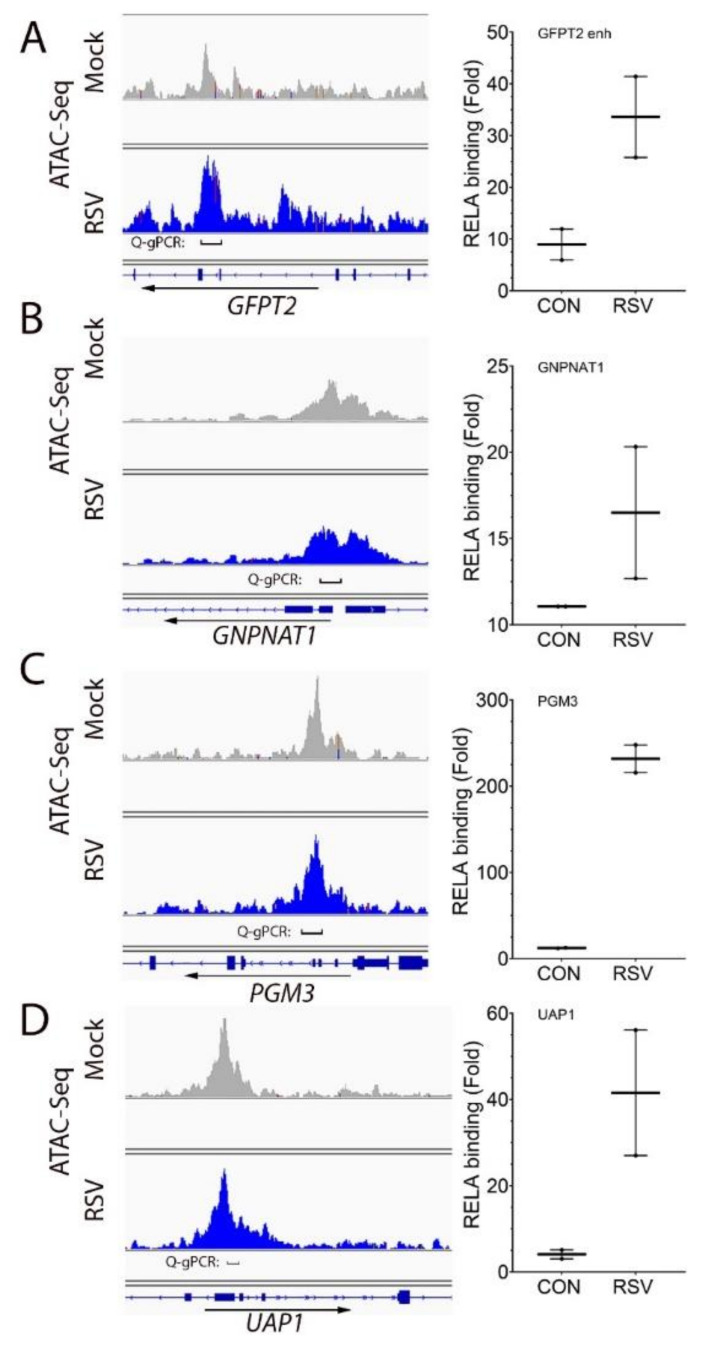
RELA binds to open chromatin domains in HBP core genes. For each gene (at left) is shown the chromatin accessibility determined by ATAC-Seq in uninfected (mock) or RSV-infected cells. The region assayed by Q-gPCR is indicated. The direction of gene transcription is indicated by arrow. At right is the Q-gPCR of mock- (CON) or RSV-infected cells analyzed by XChIP with RELA Ab. Each symbol is an independent immunoprecipitation. (**A**), *GFPT2*; (**B**), *GNPNAT1*; (**C**), *PGM3*; (**D**), *UAP1*.

**Figure 6 cells-11-02210-f006:**
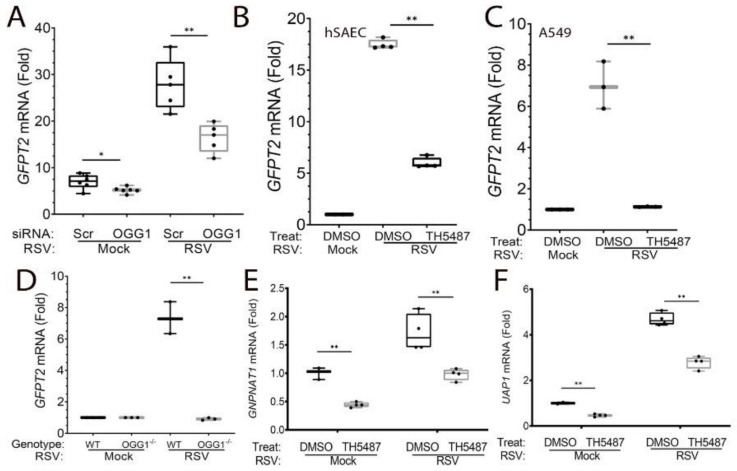
OGG1 regulates the expression of core genes in the HBP. (**A**) Scrambled (Scr) or OGG1-targeting siRNA was transfected into hSAECs. Cells were mock- or RSV-infected (MOI = 1, for 24 h), and Q-RT-PCR was performed for *GFPT2* expression. Data are presented as fold change untreated controls normalized to PPIA. (**B**) hSAECs were mock- or RSV-infected in the absence (DMSO) or presence of TH5487. Shown are the fold change of *GFPT2* mRNA. (**C**) A549 epithelial carcinoma cells were treated as in (**B**). (**D**) Wild-type or OGG1^−/−^ cells (by CrispR/Cas9) were mock-or RSV-infected. Shown is the fold change in *GFPT2* mRNA expression. (**E**) *GNPNAT1* mRNA expression in RSV-infected cells treated with solvent (DMSO) or TH5487. (**F**) *UAP1* mRNA levels in cells treated as in panel (**E**). *, *p* <0.05; **, *p* < 0.01 post-hoc Tukey’s test.

**Figure 7 cells-11-02210-f007:**
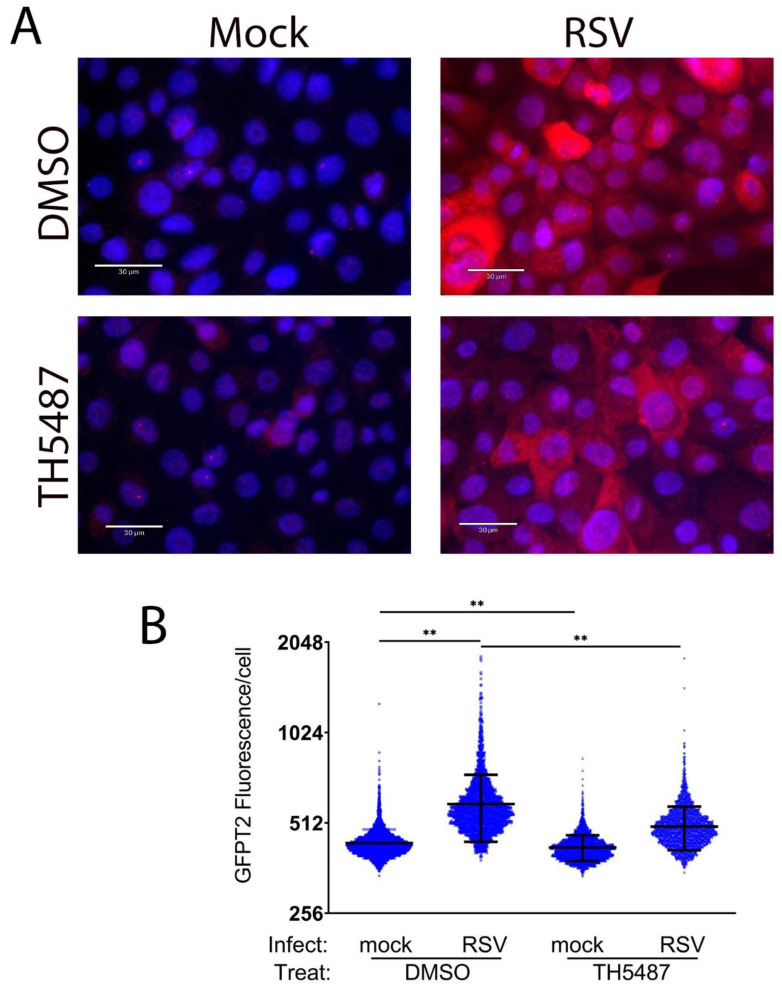
OGG1 is required for RSV-induced GFPT2 expression. (**A**) hSAECs were treated with solvent (DMSO) or TH5487 and mock- or RSV-infected as indicated. Cells were stained with anti-GFPT2 Ab (red) and nuclei by DAPI (blue). Bar indicates 30 μm. (**B**) The quantitation of GFPT2 staining. GFPT2 fluorescence was segmented by StarDist and plotted as arbitrary fluorescence/cell for each treatment condition. **, *p* < 0.01 by post-hoc Tukey comparison.

**Figure 8 cells-11-02210-f008:**
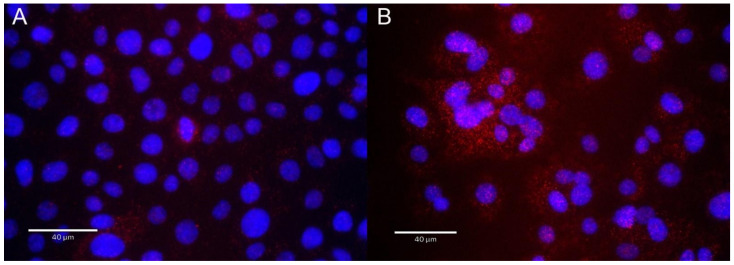
RELA∙OGG1 complex formation is induced by RSV. Shown are PLAs for RELA∙OGG1 complex in (**A**) control cells and (**B**) RSV-infected cells. Sites of protein–protein interactions (red foci) are nuclei of infected cells. Nuclei of cells are counter-stained with DAPI (blue). Note the intense staining in fused syncytia. Scale bar, 40 μm.

**Figure 9 cells-11-02210-f009:**
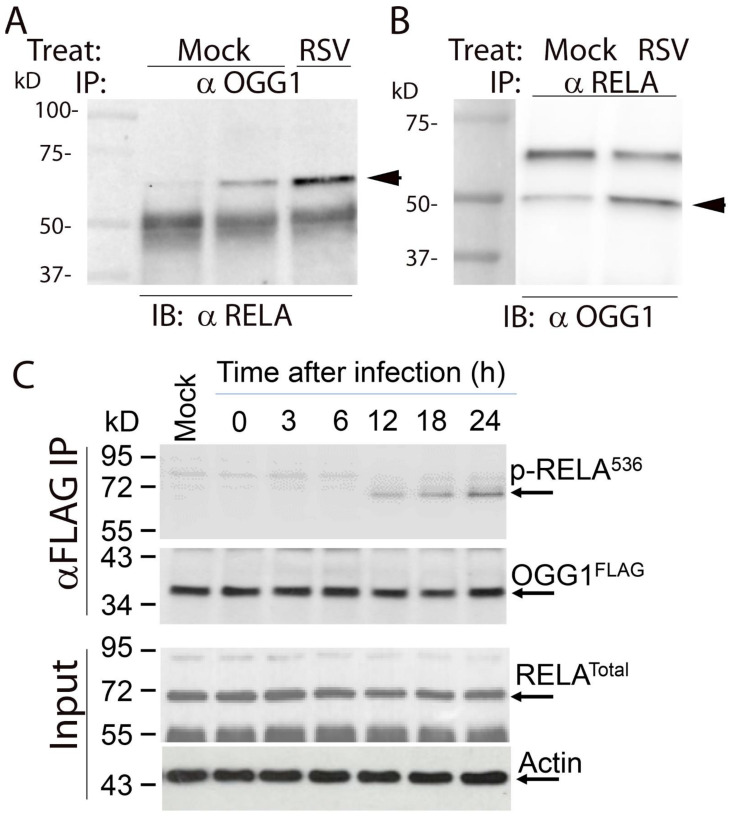
RSV induces phospho-Ser536 RELA binding to OGG1. (**A**) Co-immunoprecipitation (Co-IP) experiments. hSAECs lysates from mock- or RSV-infected cells were IPed with anti-OGG1 antibody. The immune complexes were fractionated and subjected to IB with anti-RELA. Arrowhead shows the 65 kDa RELA. (**B**) Co-IP of mock- or RSV-infected cells with anti-RELA Ab. Immune complexes were fractionated and subjected to IB with anti-OGG1. Arrowhead indicates the 50 kDa OGG1. (**C**) Lysates were prepared from FLAG-M2-OGG1-expressing A549 cells after various times of RSV infection and IPed with anti-FLAG M2 beads. Top panel, immune complexes were stained with anti-phospho-Ser536 RELA. The blot was stripped and re-probed to confirm equivalent levels of FLAG-OGG1 capture. Bottom panels are input showing total RELA and b actin. Molecular-weight markers are kDa (kD).

**Figure 10 cells-11-02210-f010:**
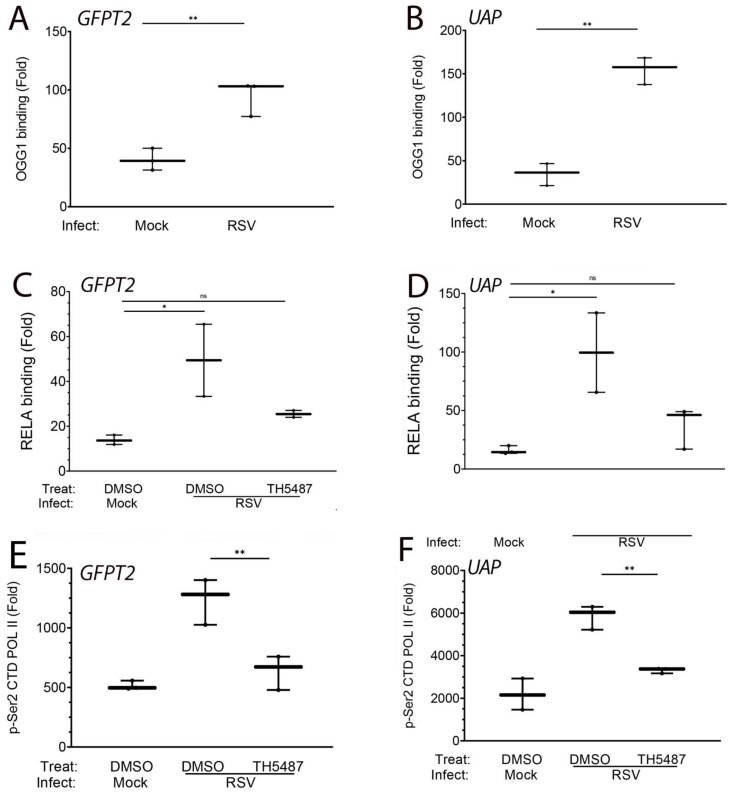
OGG1 inducibly binds to HBP regulatory elements and recruits phospho-Pol II. XChIP was performed on control or RSV-infected hSAECs; data are expressed as a fold change over IgG control. (**A**) OGG1 binding to the *GFPT2* intragenic enhancer. (**B**) OGG1 binding to the *UAP1* promoter. (**C**–**F**), Cells were RSV infected in the presence of solvent (DMSO) or TH5487 as indicated. (**C**,**D**), XChIP for RELA binding. (**E**,**F**), XChIP for phospho-Ser2 CTD Pol II binding. *, *p* < 0.05; **, *p* < 0.01; n.s., not significant. Each symbol is an independent immunoprecipitation.

**Figure 11 cells-11-02210-f011:**
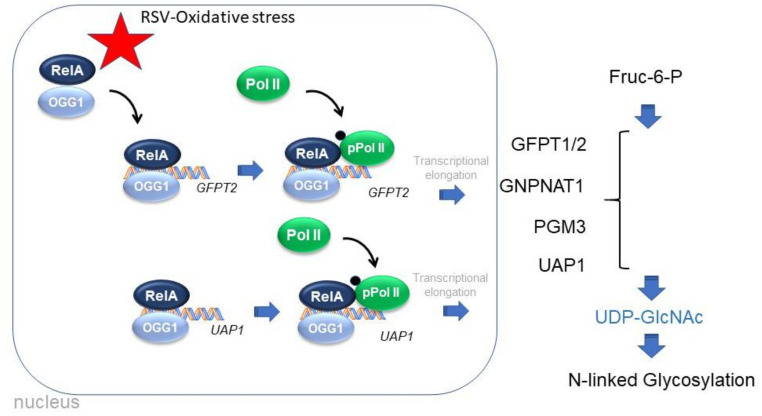
Graphical summary. Schematic of effects of RSV-induced oxidative stress on the formation of a RELAOGG1 complex that binds to nucleosome-free domains in the HBP regulatory regions. OGG1 activity is required for the RELA recruitment and association of the transcriptional elongation-competent form of phosphorylated RNA Pol II. HBP enzymes GFPT1/2, GNPNAT1, PGM3 and UAP1 catalyze the conversion of fructose-6 phosphate into UDP-GlcNAc, required for N-linked glycosylation of integrins and basal lamina proteins.

**Table 1 cells-11-02210-t001:** Quantitative RT-PCR primers.

Gene	Forward	Reverse
GFPT2	5′-GGGCATCCTGAGCGTGATTC-3′	5′-CCATGTAGCATCCCTGCTGT-3′
GNPNAT1	5′-ATCCTGGAGAAGGCTTGGTT-3′	5′-CAGAGTTGCCGTAGCAACAA-3′
PGM3	5′-TCATGTTTCGCATGGGATTA-3′	5′-AAACAGGTGGCATGTTCCTC-3′
UAP1	5′-GAGGCATTTGGAGCATTCAT-3′	5′-TCCGTCTGAGCTTCGTTTTT-3′
OGT	5′-TGTGGCAGCTTATCTTCGTG-3′	5′-GAGAGCATTGGCTAGGTTGC-3′
MGAT1	5′-GGTGGAGAAAGTGAGGACCA-3′	5′-CGGAACTGGAAGGTGACAAT-3′
OGG1	5′-GCAGCAGCTACGAGAGTCCT-3′	5′-TTCCCAGTTCCTTGTTGGTC-3′

**Table 2 cells-11-02210-t002:** Q-gPCR primers for XCHIP assay.

Genic Region	Forward	Reverse
GFPT2 enhancer	5′-GGAGTTGGGACGGAAAGTCA-3′	5′-GAAGCTCACCCTTGCCACTA-3′
GNPNAT1 promoter	5′-GGGGTAGGAGCCTAGGAAAA-3′	5′-GCGTGGGAAATGAGACAGTT-3′
PGM3 promoter	5′-GCCTAGGTCCACGTACCAGA-3′	5′-CTCGGAGTTGAGAAGGGAGA-3′
UAP1 promoter	5′-AGTGGGACAGGAGATCGTTG-3′	5′-AGAGAGGGGAAACCCAGAAA-3′

## Data Availability

ATAC-Seq studies have been previously reported and available through the GEO, accessible by the identifier: GSE161849 (https://www.ncbi.nlm.nih.gov/genbank/; accessed on 19 May 2022). The mass spectrometry proteomics data have been deposited to the ProteomeXchange Consortium via the PRIDE partner repository with the dataset identifier PXD034013 (https://www.ebi.ac.uk/pride/; accessed on 19 May 2022).

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
