# Peer review of "RELA∙8-Oxoguanine DNA Glycosylase1 Is an Epigenetic Regulatory Complex Coordinating the Hexosamine Biosynthetic Pathway in RSV Infection"

_cells, 2022, doi:10.3390/cells11142210_

Round 1
Reviewer 1 Report
I would like to thank for the opportunity to review this paper.
This paper investigated the coordination between RSV infection and the hexosamine biosynthetic pathway (HBP) through RELA and 8-Oxoguanine DNA Glycosylase1 (OGG1) in human small airway epithelial (hSAECs). First, the authors used two different RELA silencing approaches, hSAECs expressing a Doxycycline (DOX)-inducible RELA shRNA and siRNA-mediated transfection, to confirm that RSV induces the HBP in an NFκB/RELA-dependent manner. Then they used wild type and RELA-silenced cells to demonstrate that RELA is important for N glycoproteome production and the expression of core enzymes of the HBP after RSV infection. Using OGG1 inhibitor TH5487, the authors confirmed that OGG1 regulates the HBP pathway genes and will form complex with RELA during RSV infection and the OGG1 also is required for the recruitment of RELA and transcriptional elongation-competent Pol II. Those results indicate that RELA and OGG1 are epigenetic regulators for N glycoprotein and the expression of HBP pathway genes in hSAECs when infected by RSV.
This paper reported some observations which are interesting but still some parts are not well illustrated. Here are my comments:
1. This paper involves RSV infection, RELA, OGG1, and HRP pathway genes. The authors conducted some assays to clarify the relationships among them, but the details are not clear. Such as: the authors reported that RSV infection induces the expression of RELA, IL6, and HRP pathway genes, but the signal pathway is not clear. Is that RSV→RELA→OGG1→HBP pathway genes?
2. The induction of RSV infection to the expression of HBP genes can be reduced by RELA-silence or OGG1 inhibitor. But even RELA or OGG1 was blocked, compared to the mock, the RSV infected cells have higher HBP genes expression (Figure 1F, 3B, 3C, 3D, 6A, 6b, 6E, 6F, 7B …). Does this indicate that there is another signal pathway mediates RSV→?→HBP which is not RELA/OGG1 dependent?
3. In figure 4A, during RSV infection, 3 of the RELA siRNA silence cells have very strong fluorescence intensity. How to explain this?
4. The data in figure 1D, lane 2: RSV without shRNA looks weird when replicate n=3. The average value looks not match with the error bar. This also applied to Figure 1C lane 3, figure 2D bottom left image lane 2, and figure 9A lane 2, 9D lane 3.
Minor comments:
1. In Figure 1, DOX should be the same with shRNA, please label them consistently. And the Figure 1C/1E (group by mock or RSV) presentation is different from 1D/1F (group by RelA silence or not) leading to difficult understanding.
2. Lane 472 and 489, two 3.6 for the subtitle.
3. In the supplementary materials, the original western blot has one more band on the top of the image, what is that?
4. In figure 2A, the bottom right of the image has some red/yellow results in the siRNA knockdown cells when infected by RSV, why?
Author Response
Reviewer 1
Thank you for your comments. Below is a point by point response
Comment: The signal pathway is not clear.
Response: We have tried to present our findings, summarized in the final figure that RSV induces RELA to complex with OGG1, regulating the essential genes in the hexosamine biosynthetic pathway (HBP). RELA regulates some 5000 genes in the cell. IL6 is used only as an independent marker that we have blocked RELA action for the analysis of the HBP.
- Comment: Is another signal pathway mediates HBP which is not RELA/OGG1 dependent?
Response: We have earlier shown that GFPT2 is regulated independently by the unfolded protein response through XBP1s[1]. We describe in the results that this pathway partially compensates for the loss of RELA. The interaction between these two pathways is outside the scope of this manuscript.
- Comment: In figure 4A, during RSV infection, 3 of the RELA siRNA silence cells have very strong fluorescence intensity. How to explain this?
Response: XBP1s pathway partially compensates for the loss of RELA, as noted above.
- Comment: The data in figure 1D, lane 2: RSV without shRNA looks weird when replicate n=3.
Response: Results are shown as individual replicates in box plots as with horizontal line indicating median ± 10-90%. This is stated in the Methods.
Minor comments:
- We relabeled Fig 1 and its panels for consistency
- The subtitle numbering has been corrected.
- In the supplementary materials, the original western blot has one more band on the top of the image, what is that?
- In figure 2A, the bottom right of the image has some red/yellow results in the siRNA knockdown cells when infected by RSV, why?
These are a small number of peptides that increase N glycosylation in response to RELA knockdown. We do not know the mechanism.
- Qiao, D.; Skibba, M.; Xu, X.; Garofalo, R.P.; Zhao, Y.; Brasier, A.R. Paramyxovirus replication induces the hexosamine biosynthetic pathway and mesenchymal transition via the IRE1alpha-XBP1s arm of the unfolded protein response. Am J Physiol Lung Cell Mol Physiol 2021, 321, L576-L594, doi:10.1152/ajplung.00127.2021.
Reviewer 2 Report
1. SUMMARY: The manuscript “RELA∙8-Oxoguanine DNA Glycosylase1 is an Epigenetic Regulatory Complex Coordinating the Hexosamine Biosynthetic Pathway in RSV infection” by Xu et al., demonstrates how RELA∙OGG1 complex regulates hexosamine biosynthesis pathway (HBP) and impacts N-glycosylation landscape in response to RSV. Using ATACseq, glycoproteomics, and ChIP-qPCR, the authors characterize molecular mechanism of increased HBP gene expression and N-glycosylation. Overall, it is a well-written manuscript with the comprehensive cell biology assays. However, functional studies are missing (e.g., what happens when HBP pathway is perturbed during RSV infection?) and also they did not explain why GFPT2, but not GFPT1, is the main isoform of GFPT regulated by RELA-OGG1 complex (e.g., is GFPT1 regulated by other TF(s) in addition to RELA-OGG1?)
- CRITIQUE:
a. They showed IF data to claim GFPT2 expression is controlled by RELA but it would be better to do western blots to confirm their IF results. Some are not very convincing.
b. They performed PLA to show RELA form a complex with PGG but it would be nice to do IP to validate their complex formation.
c. Authors need to rectify regarding A549. They described A549 as a type II human airway epithelial cell line (ATCC CCL-185), but A549 is a lung cancer cell line not suitable to study non-cancer related diseases. Please change the description of A549.
d. As mentioned in the summary, they need to discuss why GFPT2 rather than GFPT1 is the main target regulated by RELA. At least their own speculation would be needed.
Author Response
Reviewer 2
Thank you for your comments on our manuscript. Below are point by point response to your comments
Comment: it would be better to do western blots to confirm their IF results.
Response: As noted with Reviewer 1, the XBP1 pathway is an independent pathway that partly compensates for the effect of RELA knockdown. To more convincingly demonstrate the changes and partial RELA dependence of GFPT2 expression in response to RSV, we have conducted extensive image quantification of the immunofluorescence experiments and extended these results with flow cytometry. In the first approach, we analyzed 500 independent cells for GFPT2 expression in each treatment condition using StarDist2 plug-in from FIJI. flow cytometry (Figure 4C; gating strategy is shown in Supplementary Figure S2). These data indicate that 1. RSV significantly increases cytoplasmic GFPT2 expression; 2. basal GFPT2 expression is controlled by RELA; and 3. RSV inducible GFPT2 expression is partly activated by RELA. These data are consistent with an RELA-independent mechanism for activation of GFPT2 expression.
Comment; They performed PLA to show RELA form a complex with PGG but it would be nice to do IP to validate their complex formation.
Response: We have conducted extensive IP experiments to show that RELA complexes with OGG1 and that this complex is increased in response to RSV. These data are shown as Fig. 9 in the revised manuscript.
Comment: Authors need to rectify regarding A549. .. Please change the description of A549.
Response: A549 cells are derived from adenocarcinoma as indicated by the reviewer. However, these cells are well-known to retain characteristics of type II alveolar epithelial cells, including surfactant production, aequoporin and epithelila keratin production. These cells are permissive for RSV replication and extensively used by the scientific community to study lower airway responses, since no stable type II alveolar cell line is available. We have modified the description of the A549 in the Methods to say: “A549 adenocarcinoma cells that maintain type II alveolar cell characteristics (ATCC CCL-185) are grown in minimal essential medium supplemented with 5% fetal bovine serum, streptomycin (100 ug /mL), penicillin (100 IU).”
Comment: they need to discuss why GFPT2 rather than GFPT1 is the main target regulated by RELA. At least their own speculation would be needed.
Response: We suspect that GFPT2 produces higher amounts of UDP-GlcNAc as this enzyme is less sensitive to substrate inhibition [1]. This results in higher cellular levels of UDP-GlcNAc in response to RSV.
To clarify this, in the discussion, we state: “Interestingly, our study indicates that GFPT2 is more highly inducible by RELA than the GFPT1 isoform. GFPT2 has been shown to be less sensitive to UDP-GlcNAc feedback inhibition, which may account for the higher intracellular UDP-GlcNAc levels in RSV infection.”
Reviewer 3 Report
This study studied the role of RELA·8-Oxoguanine DNA Glycosylase1 in RSV infection. They found that RELA·8-Oxoguanine DNA Glycosylase1 is an Epigenetic Regulatory Complex Coordinating the Hexosamine Biosynthetic Pathway in RSV infection. They further conclude that RSV activates the small airway epithelial HBP through transcriptional activation of 4 primary metabolic enzymes responsible for intracellular UDP-Glc-NAc accumulation. Overall, the experiments are well carried out, the data is interesting and the presentation is clear. I have a few minor questions.
- There is no full name of RELA in the Abstract.
- The text before 3.1 result maybe just move to 3.1
- All the Figures looks a bit crowd, please leave some space between panels.
- Text in Figure 2D is too small to see.
- Figure 4B, the quantification of fluorescence intensity is per cell or per view? Same as in Figure 7B.
- Both the cell outlines cannot see in Figure 4, 7 and 8.
- The left and right part in the graphic abstract in Figure 10 appears not well aligned or connected, please improve for clearance.
Author Response
Reviewer 3
Thank you for identifying these minor issues with our manuscript.
We have added the full name of the RELA in the Abstract
We moved text before Section 3.1 to 3.1 as suggested.
We separated panels in Figs 1, 4, 9, 10 for easier visibility
We have recalculated the fluorescence intensity per cell for Fig. 4B and better described the image analysis.
We have realigned the graphical abstract to eliminate the white space
Round 2
Reviewer 1 Report
The authors addressed my questions and I think it is suitable for publication.
Author Response
thank you for your comments